



# Extreme coherent gusts with direction change - probabilistic model, yaw control and wind turbine loads

Ásta Hannesdóttir, David R. Verelst, and Albert M. Urbán

DTU Wind Energy Dept., Technical University of Denmark, Roskilde, Denmark

**Correspondence:** astah@dtu.dk

**Abstract.** Observations of large coherent fluctuations from a decade of measurements are used to define a probabilistic model of coherent gusts with direction change. The gust model provides the joint description of the gust rise time, amplitude and directional changes with a 50-year return period. In conjunction with the gust model, a yaw controller is presented in this study to investigate the load implications of the joint gust variables. These loads are compared with the design load case of the extreme coherent gust with direction change (ECD) from the IEC 61400-1 Ed.4 wind turbine safety standard. Within the framework of the gust model we find the return period of the ECD to be approximately 460 years. From the simulations we find that for gusts with a relatively long rise time the blade root flapwise bending moment, for example, can be reduced by including the considered yaw controller. From the extreme load comparison of the ECD and the modeled gusts we see that by including the variability in the gust parameters the load values from the modelled gusts are between 20% and 74% higher than the IEC gusts.

## 1 Introduction

In the process of designing a wind turbine, designers have to consider a balance between cost and structural safety. Wind turbine safety standards like the IEC 61400-1 Ed.4 (IEC, 2019) exist to aid designers ensuring quality, reliability and safety of the wind turbine. The IEC standard prescribes extreme external wind conditions which the wind turbine must be able to withstand during the design lifetime, which is at minimum 20 years. The extreme wind conditions are prescribed in a set of models used for specific design load cases (DLC's). The DLC's are used to estimate the structural response to events with a recurrence period of 50 years, which is the target reliability level in wind turbine design.

The present study addresses the extreme coherent gust with direction change (ECD) which is used for DLC 1.4 for ultimate load analysis. For certain turbines this load case can drive the ultimate loading of, for example, the blade root flapwise bending moment (Beardsell et al., 2016). The ECD model is originally presented in Stork et al. (1998) and was later validated against measurements and found to give reasonable results in Hansen and Larsen (2007). As pointed out by the authors of Hansen and Larsen (2007), these experiments are based on turbulent fluctuations, where the peak values in the measurements are due to gusts with a limited spatial extent (Larsen et al., 2003). Such gusts are not coherent across the rotor diameter of multi-megawatt wind turbines, like e.g. the DTU 10 MW wind turbine (Bak et al., 2013) that we consider in this study.



In a previous study (Hannesdóttir and Kelly, 2019), observations of coherent ramp-like wind speed fluctuations are detected and characterized. The coherent fluctuations are characterized with rise time, amplitude and direction change. The observed coherent gusts are further compared with the ECD due to similarities, but show a considerable variability in the characterized variables. Generally, the rise time of the observed coherent gusts is much higher than that of the ECD, on average around 200 s. However, the rise time distribution has a large range. The observed direction change may exceed the one of the ECD, but then the corresponding rise time is considerably longer. In order to simulate a realistic wind turbine response to these observed gusts with long rise times and high direction change, yawing of the wind turbine needs to be included in the simulations.

A yaw controller ensures that the wind turbine is aligned with the mean wind speed direction. Yaw control is important for increasing the power production (Kragh and Hansen, 2015) and for reducing the extreme loading of a turbine operating in yaw misalignment. The yaw controller primarily uses the wind direction as input to determine if the turbine is operating in yaw misalignment. Conventional wind turbines have a wind vane mounted on the nacelle to calculate the yaw error. The positioning of the vane presents uncertainty on the wind direction estimate, since the equipment is installed behind the rotor where the flow is disturbed. Different sensors have been investigated for improved wind direction estimation e.g. a spinner-mounted, continuous-wave LIDAR in Kragh et al. (2010). Yaw controllers have been investigated before in connection with energy capture optimization (Bossanyi et al., 2013), but to the author's knowledge not for investigating extreme loads in conjunction with an aeroelastic code.

The aim of this paper is to investigate how 50-year coherent gusts (based on observations) impact wind turbine loads and how they compare to the DLC 1.4 of the IEC standard. This will be achieved through three steps:

1. Derive a probabilistic gust model by extrapolating the observed gust variables to a 50-year return period. As the gust variables form a three dimensional space, the extrapolation is done with the Nataf distribution model (Nataf, 1962).

2. Develop a yaw controller to incorporate in the load simulations, as the observed gusts may have a relatively long rise time, and a real wind turbine could start to yaw under such wind conditions.

3. We simulate thousands of points on the 3D gust variable surface to identify critical load regions. The load simulations are performed using the aeroelastic software HAWC2 (Larsen and Hansen, 2015).

## 2   The IEC extreme coherent gust with direction change

The amplitude of the extreme coherent gust with direction change (ECD) is $V_{cg} = 15$ m/s and is independent of the 10-minute mean hub height wind speed $V_{hub}$. The direction change of the ECD is however a function of $V_{hub}$ and given by

$$\theta_{cg} = \begin{cases} 180°, & \text{if } V_{hub} \leq 4 \text{ m/s}, \\ 720° \, (\text{m/s})/V_{hub}, & \text{if } 4 \text{ m/s} < V_{hub} < V_{ref}, \end{cases} \tag{1}$$



where $V_{\text{ref}}$ is the 10-minute mean reference wind speed. The wind speed increase and direction change are assumed to occur simultaneously and are modeled as functions of time,

$$
V(z,t) = \begin{cases} V(z), & \text{if } t < 0 \\ V(z) + 0.5 V_{\text{cg}}(1 - \cos(\pi t/T)), & \text{if } 0 \leq t \leq T \\ V(z) + V_{\text{cg}}, & \text{if } t > T \end{cases} \tag{2}
$$

$$
\theta_{\text{cg}}(t) = \begin{cases} 0°, & \text{if } t < 0 \\ \pm 0.5 \theta_{\text{cg}}(1 - \cos(\pi t/T)), & \text{if } 0 \leq t \leq T \\ \pm \theta_{\text{cg}}, & \text{if } t > T \end{cases} \tag{3}
$$

where $T = 10\,s$ is the rise time. The ECD design load case is simulated at three different wind speeds, $V_{\text{rated}}$ (the rated wind speed) and $V_{\text{rated}} \pm 2$ m/s according to the IEC standard. In this study we only simulate the ECD with start wind speed of 10 m/s.

## 3 Probabilistic model of coherent gusts

In the present study we consider wind speed fluctuations that may be assumed to be coherent across the rotor of any multi-megawatt wind turbine. Such large coherent fluctuations are detected and characterized in a previous study (Hannesdóttir and Kelly, 2019), where a detailed description of the detection and characterization method may be found. The coherent gusts are detected from a 10.25 year measurement period in Høvsøre, located in Jutland, Denmark. It is argued that these gusts may originate from a broad variety of phenomena, but share the trait of ramp-like increase in wind speed that gives rise to extreme 10-minute variance.

The events are characterized by rise time ($\Delta t$), direction change ($\Delta \theta$) and amplitude ($\Delta u = u_a - u_b$), where $u_b$ is the wind speed before the rise, and $u_a$ is the wind speed after the rise. The events are observed at wind speeds ranging from $u_b = 1.4$ m/s to $u_b = 26.4$ m/s. In order to model the gusts, we choose a subset of events with the following selection criteria: $u_b < V_{\text{rated}}$ and $u_a > V_{\text{rated}}$. In other words, the wind speed is below rated wind speed before the gust and reaches above rated wind speed after the gust. This choice of subset is made as high loads are expected to be observed around rated wind speed (Hannesdóttir et al., 2019). A total of 92 events fulfill the selection criteria, and are used in the present study.

Figure 1 shows the cumulative distribution (CDF) of each coherent gust variable $\Delta u$, $\Delta \theta$ and $\Delta t$ with fitted distributions. The distribution parameters are all found with maximum likelihood estimation. The gust amplitude $\Delta u$ is assumed to follow a Gumbel distribution, where the estimated location and scale parameters are $\alpha = 6.42\,\text{m/s}$ and $\beta = 1.77\,\text{m/s}$. The direction change $\Delta \theta$ is assumed to follow a three-parameter Weibull distribution with estimated parameters: $k = 1.34$ (shape parameter), $\gamma = 6.37°$ (location parameter) and $A = 25.30°$ (scale parameter). The rise time is assumed to follow the reversed two-parameter Weibull distribution, where the estimated parameters are: $k = 1.47$ and $A = 279.37\,\text{s}$. Note that we change the sign of the rise time when fitting the reversed Weibull distribution. This is done to ensure that the short rise times are defined as extreme values.

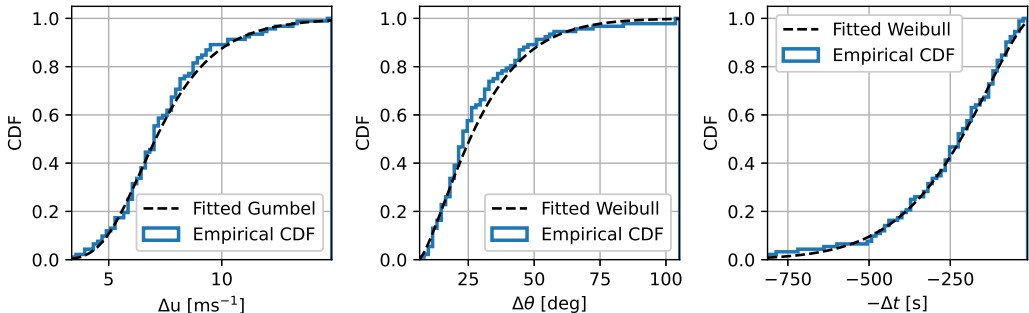

**Figure 1.** The marginal fitted and empirical cumulative distributions of: gust amplitude (left), gust direction change (middle) and negative gust rise time (right).

### 3.1 Inverse second-order reliability method with the Nataf distribution model

In this section we derive an environmental surface that provides the 50-year return period of the joint coherent gust variables, $\Delta u$, $\Delta \theta$ and $\Delta t$. Traditionally, the inverse first order reliability method (IFORM) (Winterstein et al., 1993) is used within wind energy to predict such extreme environmental conditions, e.g. for the 50-year return period of the joint description of wind speed and turbulence levels (Fitzwater et al., 2003; Sang Moon et al., 2014; Dimitrov et al., 2017). The IFORM has however recently been shown to underestimate the exceedance probability by an order of magnitude, leading to a non-conservative return period prediction (Chai and Leira, 2018; Dimitrov, 2020). We therefore use the inverse second-order reliability method (ISORM), which is a method developed by Chai and Leira (2018) to replace the IFORM. The ISORM method follows the same general procedure and steps of the IFORM, except for when the so called 'reliability index' is calculated (see equation 7 below), where the ISORM provides an exact solution for the return period.

Here the joint probability distribution of the variables is given by the Nataf distribution model, which is defined by Liu and Der Kiureghian (1986). Under Nataf transformation (Nataf, 1962), the considered variables are mapped from original space into correlated standard normal space, where the joint description of the variables is defined by a Gaussian copula. Unlike the Rosenblatt transformation (Rosenblatt, 1952), which is exact, the Nataf transformation is an approximate. In order to perform the Rosenblatt transformation, the complete joint cumulative distribution of the variables is needed, which is not available for the current analysis. However, to perform the Nataf transformation it is enough to know the marginal distributions and the correlation matrix of the variables.

The Nataf transformation of a random vector $\mathbf{X} = X_1, ..., X_n$ to standard normal space is performed by

$$Z_i = \Phi^{-1}(F_{x_i}(X_i)), \qquad i = 1, ..., n \tag{4}$$

where $\Phi^{-1}$ is the inverse standard CDF and $F_{x_i}(x_i)$ is the marginal CDF of $X_i$. The standard normal vector $\mathbf{Z} = Z_1, ..., Z_n$ has a correlation matrix $\boldsymbol{\rho}_0$.



### 3.1.1 Constructing the environmental surface

The first step in constructing the environmental surface of coherent gust variables is to calculate the exceedance probability ($P_e$) associated with the 50-year return period of an observed coherent gust. When the data has a regular time step, the probability of a single 10-minute measurement to exceed the 50-year value is $P_{50} = T_0/T_{50}$, where $T_0$ is a 10-minute reference period and $T_{50}$ is the 50-year return period with the same unit as the reference period. However, as the coherent gusts do not occur with a regular time step, the exceedance probability has to be adjusted with the annual exceedance frequency of the detected coherent gusts $f_a = n_{cg} \cdot T_0/T_m$, where $n_{cg}$ is the total number of observed coherent gusts and $T_m$ is the measurement period converted to the same unit as the reference period. There are 92 selected coherent gusts found from 10-minute samples spanning a 10.25 year period, giving the exceedance probability:

$$P_e = \frac{P_{50}}{f_a} = \frac{1}{50 \cdot (92/10.25)} = 0.0022 \tag{5}$$

The next step is to find the associated 'reliability index', which has its name from the traditional first order reliability method (Ditlevsen and Madsen, 1996). Following the IDS methodology, the reliability index is defined by

$$\beta = \sqrt{D_k^{-1}(1 - P_e)} = 3.82 \tag{6}$$

where $D_k^{-1}$ is the inverse CDF of the chi-square distribution with $k$ degrees of freedom. Here we use $k = 3$ because we have a three dimensional gust variable space. In standard normal space $\beta$ defines the radius of a sphere[1]

$$\beta = \sqrt{u_1^2 + u_2^2 + u_3^2} \tag{7}$$

where $u_1$, $u_2$ and $u_3$ are spherical coordinates of the vector $|\mathbf{U}| = \beta$. The spherical coordinates may be generated by

$$u_1 = \beta \cos(\theta) \sin(\phi) \tag{8}$$
$$u_2 = \beta \sin(\theta) \sin(\phi) \tag{9}$$
$$u_3 = \beta \cos(\phi) \tag{10}$$

where $\theta = [0, 2\pi]$ and $\phi = [0, \pi]$.

Before performing the Nataf transformation, the correlation coefficients of $\boldsymbol{\rho}_0$ have to be determined. As shown in Liu and Der Kiureghian (1986), the correlation coefficients in standard normal space can be estimated from the correlation coefficients $\rho_{ij}$ in real space ($\Delta u$, $\Delta \theta$ and $\Delta t$), through the following expression:

$$\rho_{0ij} = E \, \rho_{ij} \tag{11}$$

where $E \geq 1$, and is a function of the correlation coefficient $\rho_{ij}$ and the corresponding marginal distributions. Empirical expressions for $E$ are provided in Liu and Der Kiureghian (1986) for 10 different distribution functions, where the Weibull

---

[1]Note that the chi-square distribution is by definition the distribution of a sum of the squares of $k$ independent standard normal random variables. It can therefore be seen from eq. 7 that using the chi-square CDF in eq. 6 is theoretically correct.



| ij | $\rho_{ij}$ | $E(\rho_{ij})$ | $\rho_{0ij}$ |
|---|---|---|---|
| $\Delta u \; \Delta \theta$ | 0.498 | 1.070 | 0.534 |
| $\Delta u \; \Delta t$ | -0.292 | 1.114 | -0.325 |
| $\Delta \theta \; \Delta t$ | -0.296 | 1.069 | -0.316 |

**Table 1.** The estimated correlation coefficients and the evaluated empirical expressions for $E$.

distribution and the Gumbel distribution are both among them. We can therefore use these empirical expressions to estimate $E$ (see Table 1). The correlation matrix $\boldsymbol{\rho}$ is calculated for the variables $\Delta u$, $\Delta \theta$ and $\Delta t$, and eq. 11 is used to estimate the correlation coefficients of $\boldsymbol{\rho}_0$ (see Table 1).

We can now determine $\boldsymbol{\rho}_0$ as a lower-triangular matrix $\mathbf{L}_0$ by applying Cholesky decomposition. The last step in constructing the environmental surface of coherent gust variables, is to apply the inverse Nataf transformation. This is done in two steps. First to transform $\mathbf{U}$ to the correlated standard normal space,

$$\mathbf{Z} = \mathbf{L}_0 \cdot \mathbf{U} \tag{12}$$

and finally the variables of the surface are found by

$\Delta u = F_{\Delta u}^{-1}(\Phi(Z_1))$                             (13)

$\Delta \theta = F_{\Delta \theta}^{-1}(\Phi(Z_2))$                              (14)

$\Delta t = F_{\Delta t}^{-1}(\Phi(Z_3)).$                               (15)

     Figure 2 shows the surface of coherent gust variables with a 50-year return period and the 92 coherent gust events used in this analysis. The maximum direction change ($\Delta \theta = 143.0°$) and maximum amplitude ($\Delta u = 23.5$ m/s) are found on the

surface at rise times of 479.4 s and 471.0 s respectively. The surface is shown as $\Delta u$ - $\Delta \theta$ contours for specific rise times in Figure 3. It may be seen that extreme direction change and amplitude decrease with decreasing rise time. The grey curve shows the $\Delta u$ - $\Delta \theta$ contour where the rise time matches the rise time of the ECD $\Delta t = 10$ s. On that curve we find the maximum $\Delta u = 13.1$ m/s and the maximum $\Delta \theta = 75.4°$.

### 3.1.2    The return period of the ECD

We see from the gust variables on the 10 s rise time contour in Figure 3, that if we match the direction change of the ECD ($\Delta \theta = 72°$) the amplitude is significantly lower, $\Delta u = 10.0$ m/s. One could then ask: What is the required return period of the ECD gust parameters when considered within this framework? Or in other words, what return period do we have to use in



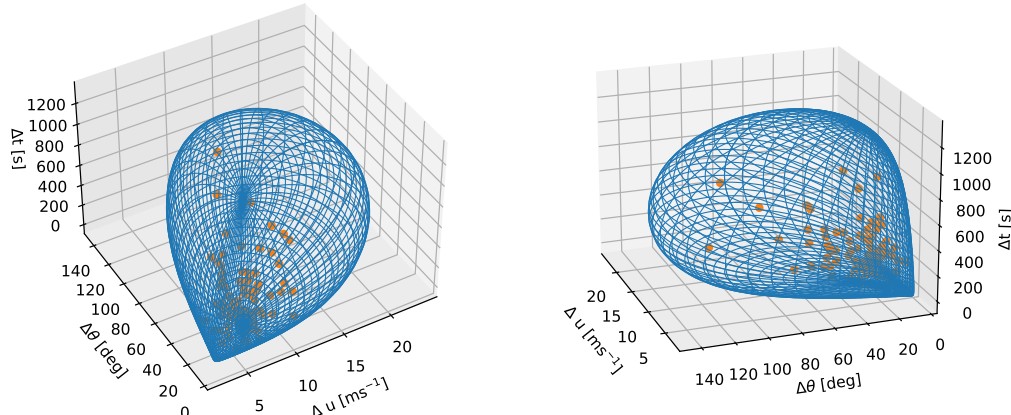

**Figure 2.** The 50-year return period surface of $\Delta u$, $\Delta \theta$ and $\Delta t$, seen from two different angles and the 92 coherent gust events.

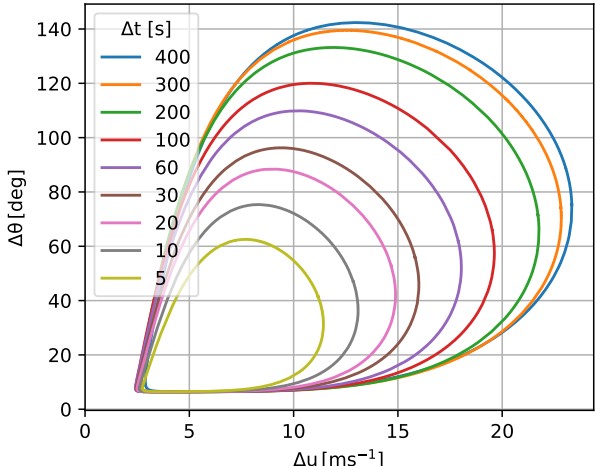

**Figure 3.** The 50-year return period surface, sliced at different rise times.

order to make a point on the surface match all three ECD variable values? The answer to that question may be found by doing the reversed operation of constructing the environmental surface:

$$
\quad \mathbf{Z}_{\text{ECD}} =
\begin{cases}
\Phi^{-1}(F_{\Delta u}(15)) \\
\Phi^{-1}(F_{\Delta \theta}(72)) \\
\Phi^{-1}(F_{\Delta t}(-10))
\end{cases}
\tag{16}
$$



Followed by the step

$$\mathbf{U}_{\mathrm{ECD}} = \mathbf{L}_0^{-1} \cdot \mathbf{Z}_{\mathrm{ECD}}, \tag{17}$$

The reliability index may be found by $\beta_{\mathrm{ECD}} = |\mathbf{U}_{\mathrm{ECD}}|$ and the associated probability by $P_{\mathrm{ECD}} = 1 - D_k(\beta_{\mathrm{ECD}}^2)$. Finally the return period is found

$$T_{\mathrm{ECD}} = \frac{1}{P_{\mathrm{ECD}} \cdot 92/10.25\,\mathrm{years}} = 460.4\,\mathrm{years} \tag{18}$$

A return period of 460 years is an order of magnitude larger than the usual 50-year return period used in wind turbine design. The reason for this large return period is that according to our distributions and correlations between the coherent gust variables, there is a low probability that a coherent gust with a rise time of 10 seconds has such a large amplitude as the ECD. Another reason for this large return period is that the ECD is originally based on point measurements of turbulent fluctuations, as

mentioned in the introduction. These fluctuations generally have short time scales and high peak values, but are not necessarily coherent.

## 4 Aeroelastic simulation environment

In this section we briefly describe the HAWC2 software that was used for performing aeroelastic simulations and the HAWC2 yaw controller that was specially developed for this study.

### 4.1 HAWC2

HAWC2 (Larsen and Hansen, 2015) version 12.8 is used to calculate the aero-servo-elastic response of the DTU10MW (Bak et al., 2013). The DTU10MW is used together with the open source Basic DTU Wind Energy Controller (Hansen and Henriksen, 2013). The source code can be found at (Hansen and Tibaldi, 2018). As reference load case the IEC DLC 1.4 ECD is used at 10 m/s. From a blade element momentum (BEM) modelling point of view, the considered load cases here are affected

in particular by the dynamic inflow model (gust rise time and amplitude), the induction correction due to skewed inflow (yaw misalignment), and the non-constant induction along an annular ring element. These corrections, and other HAWC2 BEM modelling specifics, are discussed in (Madsen et al., 2020).

### 4.2 Yaw controller

When a yaw error is measured, different strategies are used to determine whether the wind turbine must yaw for a given

misalignment. The yaw controller presented in Kragh et al. (2013) uses a periodic correction of the yaw angle where the misalignment error is low pass filtered and integrated. Once the integrated error exceeds the defined threshold, the wind turbine starts yawing. The yaw controller designed for this study instead uses two moving averages of the misalignment error, of different length. The first moving average is used to determine the initialization of the yawing sequence while the second one commands the stop. The yaw controller is used along with the Basic DTU Wind Energy Controller (Hansen and Henriksen,





185 2013).

The two dimensional wind vector, $\boldsymbol{V}_{\text{hub}}$, is used as input for the yaw controller, where the instantaneous yaw error, $\theta$, is calculated from the lateral and longitudinal components ($V_x$ and $V_y$) of $\boldsymbol{V}_{\text{hub}}$. Further, the yaw controller needs some additional user-defined parameters; the length of the average window for yaw start ($m$), the length of the average window for yaw stop ($n$), a yaw-error threshold for start ($\theta_t$) and a yaw-error threshold for stop ($\theta_s$).

The moving average, used to trigger the initiation of the yaw sequence, is computed as

$$\alpha(t) = \frac{1}{m} \sum_{i=0}^{m-1} \theta(t - i\,dt), \tag{19}$$

where $t$ is the current time and $dt$ is the simulation frequency.

Similarly, the moving average used to stop the yaw action is defined

$$\beta(t) = \frac{1}{n} \sum_{i=0}^{n-1} \theta(t - i \cdot dt). \tag{20}$$

To ensure a proper function of the simple yaw controller, the condition $m < n$ needs to be imposed. Additionally, the yaw-error thresholds need the condition $\theta_t > \theta_s$, which here is fulfilled by setting the threshold stop as $\theta_s = (1/2)\theta_t$.

The wind turbine starts yawing once $\alpha(t)$ exceeds the yaw error threshold, $\theta_t$. The operation mode of the yaw action can either be on, or off, and is here defined

$$\epsilon(t) = \begin{cases} 1, & \text{if } |\alpha(t)| > \theta_t \\ 0, & \text{if } |\beta(t)| \leq \theta_s \end{cases} \tag{21}$$

where 1 corresponds to active yawing and 0 to inactive yawing. The yawing sequence stops once $\beta$ is lower than the yaw-stop threshold $\theta_s$. The yaw controller imposes the yawing rate, $\gamma$, and its direction by

$$\gamma(t) = \begin{cases} +\gamma_{max}, & \text{if } \alpha(t) > \theta_t \\ -\gamma_{max}, & \text{if } \alpha(t) < -\theta_t \\ 0, & \text{if } \beta(t) < \theta_s \end{cases} \tag{22}$$

In this case, the yawing rate is always chosen as the maximum, $\pm\gamma_{max}$. After the yaw sequence stops, the operation mode is

defined as $\epsilon(t) = 0$, and the computation of both moving averages, $\alpha(t)$ and $\beta(t)$ is initialized.

Figure 4 illustrates an idealized example of the yaw controller action, where the nacelle is initially offset 30 degrees with respect to the mean wind direction. The stop moving average $\beta(t)$ (red line), with a time window of 10 seconds, responds much faster to a change in yawing error compared to the start moving average $\alpha(t)$ (blue line) with a window length of 120 seconds. The threshold $\theta_t$ is set to 5 degrees and it can be observed how the wind turbine starts yawing once $\alpha(t)$ exceeds the

threshold (right blue marker). The turbine will continue yawing until $\beta(t)$ (red line) is below $\theta_s$ (right red marker), where the instantaneous yaw error is 0.2 degrees.

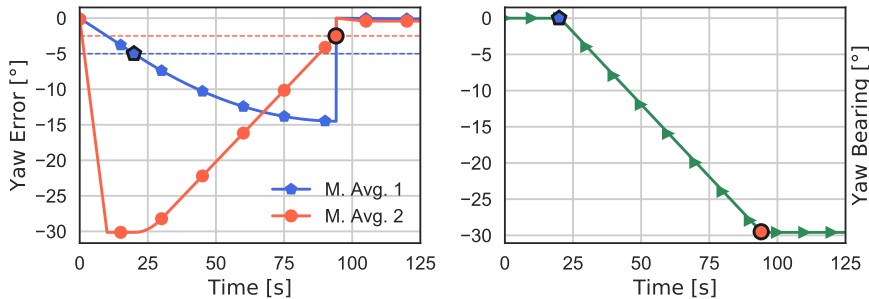

**Figure 4.** *Left*: Yaw error as function of time for; start moving average 1 (blue), and stop moving average 2 (red). The blue dashed line shows the start threshold and the red dashed line shows the stop threshold. *Right*: Yaw bearing angle as function of time.

Although a more elaborate study regarding yaw control should be considered, the simple long/short time averaging approach is chosen here in order not to trigger a yaw action too early (hence 120 second averaging window for the start trigger), while avoiding to overshoot after a zero yaw error has been reached (using the 10 second averaging window for the stop trigger).

The yaw mechanism is modelled as a second order dynamical system with a frequency of 5 Hz and a damping ratio of 0.7. There is no limit on the maximum and minimum yaw angle allowing a full rotation of the system. It is possible to constrain the response of the second order model in velocity and acceleration. A typical yaw sweep sequence, yawing 360 degrees, lasts around 15 minutes which leads to a yaw rate of approximately 0.4 °/s. The proposed basic yaw controller can be replaced with a Proportional Integral Derivative (PID) controller where the objective signal is the yaw angle and not the rate of change or

velocity of the yaw mechanism.

## 5 Simulation results

This section contains the results from the HAWC2 simulations. The simulations consist of 3219 points on the 50-year joint gust variable surface, with different rise times, amplitudes, and direction changes. The gust surface has been sliced to limit the presentation of simulation results to relevant ranges. The simulation results are shown for a rise time range of $4 - 400$ s, where

the lower rise-time threshold for this range is chosen from considerations of the shortest possible fluctuation turnover time where the wind field can be considered coherent across the rotor of the DTU 10 MW wind turbine. Further, when considering curves with the same $\Delta t$ on the 3D gust variable surface, only higher values of the combination of $\Delta\theta$ and $\Delta u$ are shown for the simulation results (values closest to the origin in Figure 3 are filtered out).

We have simulated the IEC ECD coherent gust for comparison. All simulations have been performed both with and without

yaw control and all gust simulations, including the IEC simulation, start from 10 m/s. We discuss the implication of using a yaw controller and give an overview of all the simulated gusts and corresponding load maxima. Among the load channels we investigate in this section, we analyse the tower-bottom resultant bending moment ($TB_{res}$). This binding moment is defined as $TB_{res} = \sqrt{TB_{FA}^2 + TB_{SS}^2}$, where $TB_{FA}$ is the tower base fore-aft moment and $TB_{SS}$ is the tower base side-side moment.



The tower-base resultant bending moment provides a clear presentation of the total magnitude of the tower bottom loading, with and without yawing.

## 5.1 Load response on the 3D gust surface

In this section we focus on the load simulations that were performed with the yaw controller, as we believe that these simulations best represent load response of a real operating wind turbine.

Figure 5 and 6 give a global overview of the tower bottom resultant ($TB_{res}$), tower top yaw ($TT_{yaw}$), blade root flap ($BR_{flap}$), and blade root edge-wise ($BR_{edge}$) loads. The colour indicates the absolute maximum load response of each simulation, which is shown as a dot on the surface. Due to the complexity of the considered 3D gust domain, two alternative views are given to illustrate the response: for Figure 5 the 3D gust surface is used, while for Figure 6 the gust amplitude and rise time axis are collapsed into a gust acceleration dimension (amplitude divided by rise time), and the gust acceleration is plotted on a logarithmic scale.

In Figure 5 a bright area (indicating the highest loads) for the tower base and blade root bending moments may be seen. This area corresponds to gusts with low direction changes, short rise times, and relatively modest amplitudes. This is the area for which the absolute maxima of the loads occurs, as can also be seen in Table 2. In Figure 6 this area of highest blade root and tower base loads is seen for the highest gust accelerations. Further we notice in the highest gust accelerations area for the tower base and blade root edge loads, that for increasing direction change the max load responses decrease. These load channels are highly connected with the rotor thrust, that decreases by introducing moderate direction change.

The blade-root flap-wise moments also show a circular shape on the surface with moderately high load responses for a large range of the gust variables on the surface (Figure 5). For this area of the gust surface the direction change is high and high yaw errors occur, as the yaw controller reacts slower than the direction change of the gusts occur. It has been verified from the simulations in this area that the rotor speed drops to the minimum after the gust passage, due to the high yaw errors (approaching 90 degrees). After the gust passage the yaw controller starts to align the turbine again with the wind, and which is now well above rated wind speed. The peak blade root bending moments occur around the time the turbine when the pith controller becomes active again to reduce the rapidly increasing aerodynamic thrust and power due to the decreasing yaw angle.

For the tower top yawing moment a large area (on the right of the 3D gust domain) with high loading corresponds to gusts with high amplitudes, wide range of high rise times, and large changes in direction. The yaw controller is active when the absolute highest tower top yaw moment occur in the simulations, and the shape of the high load area is highly influenced by the current implementation of the yaw controller.

## 5.2 Comparison with the IEC ECD

The absolute maxima of the response of selected load channels over the considered gust surface is compared with the IEC ECD definition in Table 2. When looking at the blade and tower loading it is observed that IEC ECD gust is not conservative, and that especially gusts with a lower amplitude and shorter rise time (compared to the IEC ECD) impose a tougher loading



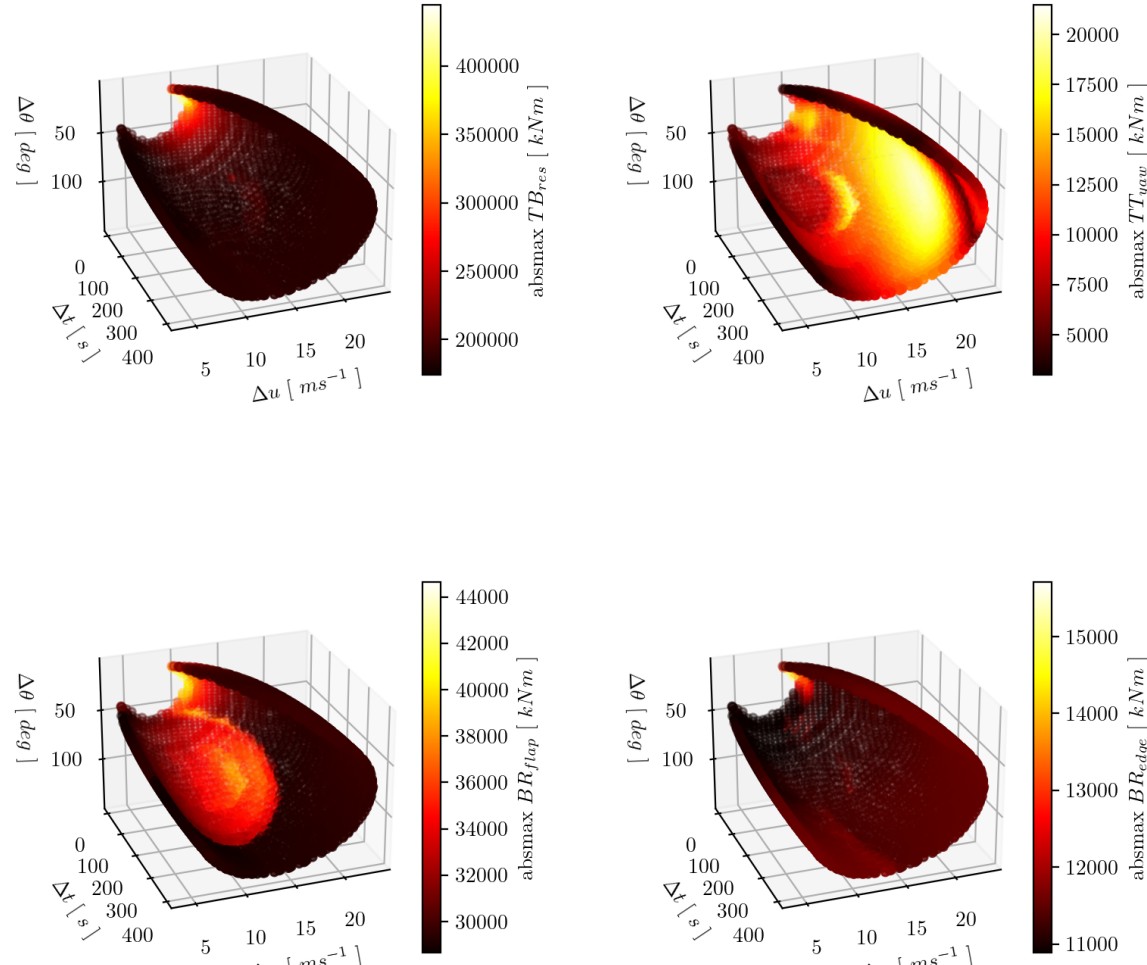

**Figure 5.** Simulation results with yaw control. The absolute maximum responses are shown with the colour scale on the 3D gust variable surface of rise time, amplitude and direction change.

condition. Further, it can be noted that all extreme loads for tower base and blade root bending moments occur for very similar gusts: amplitude of 10.3 or 10.7 m/s and with rise times between 4.0 and 4.4 seconds. Note that for these short gusts one could also consider the initial azimuth position of the rotor to further assure the most critical condition is selected. The tower top

torsional loading (in yawing direction) is different compared to the other channels. The maximum for the $TT_{yaw}$ occurs for a gust with long rise time, high amplitude and large direction change.



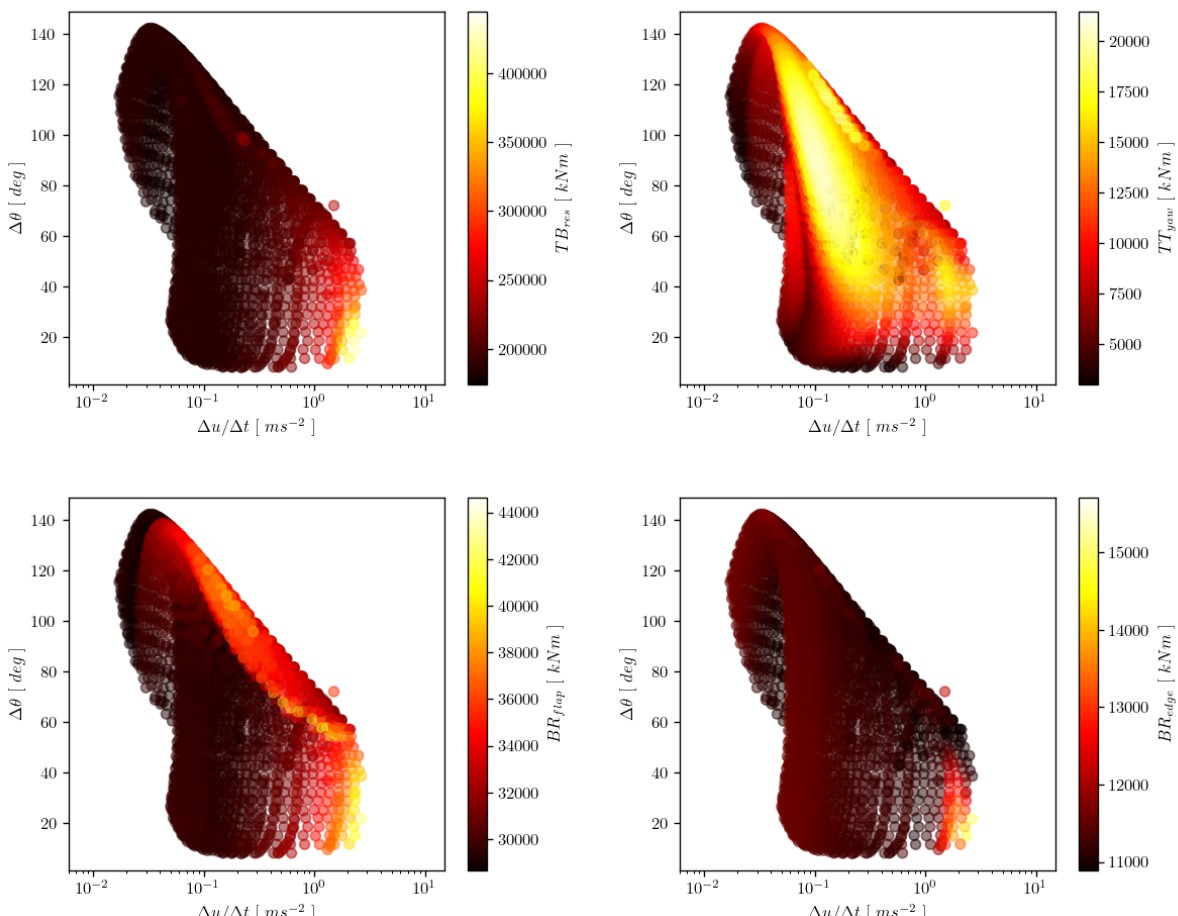

**Figure 6.** Simulation results with yaw control. The absolute maximum responses are shown with the colour scale on the 2D gust variable domain of acceleration and direction change. Notice the IEC ECD gust on the right panel is the single outlier.

| Ampl. [ms$^{-1}$] | Rise time [s] | Acc. [ms$^{-2}$] | Dir. change [°] | channel | absmax [kNm] | IEC absmax [kNm] | % diff wrt IEC |
|---|---|---|---|---|---|---|---|
| 10.3 | 4.4 | 2.34 | 14.9 | $TB_{res}$ | 423572.1 | 243120.1 | 74.2 |
| 21.0 | 186.2 | 0.11 | 85.3 | $TT_{yaw}$ | 20459.0 | 16598.7 | 23.3 |
| 10.7 | 4.0 | 2.66 | 21.6 | $BR_{flap}$ | 42522.2 | 35562.8 | 19.6 |
| 10.7 | 4.0 | 2.66 | 21.6 | $BR_{edge}$ | 14963.5 | 12121.6 | 23.4 |
| 10.7 | 4.0 | 2.66 | 21.6 | $BR_{torsion}$ | 508.9 | 310.5 | 63.9 |

**Table 2.** Absolute maxima of a selection of load channels with yaw control for gusts starting at 10 m/s. The IEC absmax column refers to the IEC DLC1.4 gust case with a rise time of 10 seconds, amplitude of 15 m/s and a directional change of $72°$.



The difference between what drives the tower top yawing moment and the tower base resultant moments is illustrated in Figure 7. The magnitude of the load is given on the ordinate (y-axis) and the gust acceleration is plotted on the abscissa (x-axis). The colors mark the magnitude of the direction change of the gust and the IEC ECD load case is indicated with a white

cross (and further marked with a dashed horizontal line). We see that the tower base loads are driven by the gust acceleration, with almost a linear relationship between gust acceleration and the load response (for similar direction change). It can also be noticed that for comparable gust accelerations, the tower base resultant loads are higher when the direction change is lower. The gust acceleration influences the tower top yawing moments to a lesser extent. The absolute maxima occur for very slow gusts with very large direction changes.

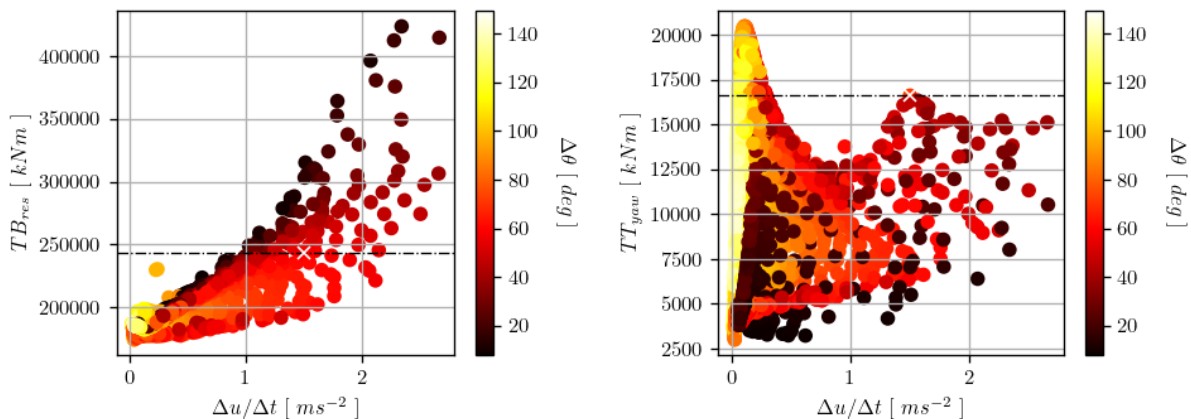

**Figure 7.** Simulations with yaw control. *Left panel*: Tower base resultant moment as function of gust acceleration. The colors of the markers refer to the gust directional change, in degrees. *Right panel*: Tower top yawing moment as function of gust acceleration. The marker colors refer to the gust direction change in degrees. The IEC ECD case is marked with a white cross and its value on the y-axis is indicated with the horizontal dashed line.

An example time series of the gust resulting in the highest tower bottom resultant load and tower top acceleration is compared with the IEC ECD gust in Figure A1 in Appendix A. Additionally, the gust resulting in the highest tower top yawing moment compared with the IEC ECD gust is shown in Figure A2.

### 5.3 The effect of the yaw controller

The simulations on the gust variable surface were performed both with and without the yaw controller. This was done to

investigate the differences in load response between the two simulation sets. Table 3 shows the absolute maxima from a selection of load channels from the HAWC2 simulations that were run without a yaw controller. When comparing the results shown in Table 3 and Table 2 it can be seen that the absolute maximum loads are all reduced by including the yaw controller.

In Figure 8 the absolute maximum of the blade-root flap-wise bending moment $BR_{flap}$ is considered. In the left panel we see the results for the simulations that were run without the yaw controller and in the right panel the results that were run with



| Ampl. [ms$^{-1}$] | Rise time [s] | Acc. [ms$^{-2}$] | Dir. change [°] | channel | absmax [kNm] | IEC absmax [kNm] | % diff wrt IEC |
|---|---|---|---|---|---|---|---|
| 10.3 | 4.4 | 2.34 | 14.9 | $TB_{res}$ | 423583.7 | 243109.8 | 74.2 |
| 21.3 | 381.8 | 0.06 | 111.8 | $TT_{yaw}$ | 22363.4 | 16598.9 | 34.7 |
| 20.4 | 396.9 | 0.05 | 119.6 | $BR_{flap}$ | 45581.5 | 33887.1 | 34.5 |
| 22.5 | 324.7 | 0.07 | 90.7 | $BR_{edge}$ | 15742.4 | 12121.6 | 29.9 |
| 20.4 | 396.9 | 0.05 | 119.6 | $BR_{torsion}$ | 818.9 | 310.5 | 163.7 |

**Table 3.** Absolute maxima of a selection of load channels without yaw control for gusts starting at 10 m/s. The IEC absmax column refers to the IEC DLC1.4 gust case with a rise time of 10 seconds, amplitude of 15 m/s and a directional change of 72°.

the yaw controller. For this load channel there is a clear dependency on the presence of a yaw controller and we see that the absolute maximum loads are reduced by including yaw control. This reduction is for gusts with a long rise time.

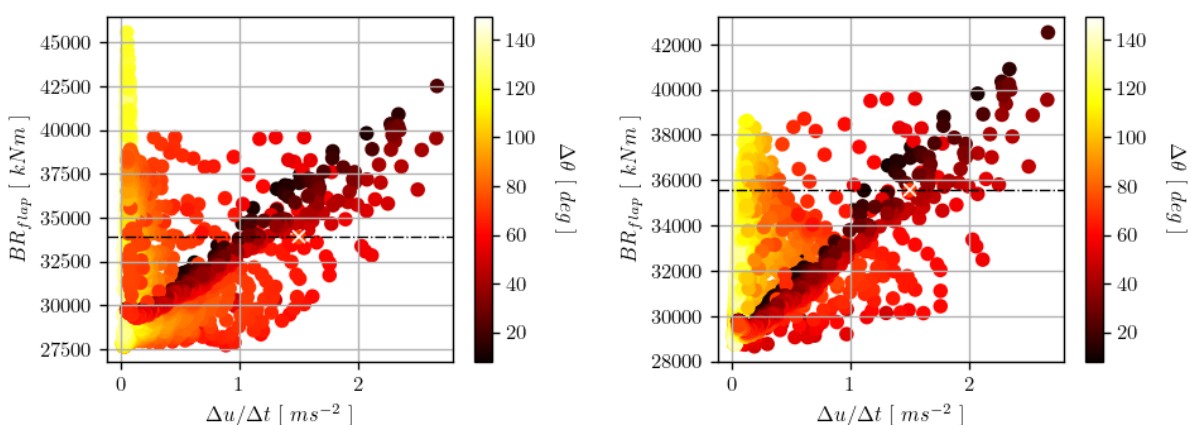

**Figure 8.** Simulations without yaw control (left panel), and with yaw control (right panel). Blade root flap-wise loads as function of gust acceleration. The colors of the markers refer to the gust directional change, in degrees. The IEC ECD case is marked with a white cross and its value on the y-axis is indicated with the horizontal dashed line.

## 6 Discussion

In this analysis we chose to extrapolate the multivariate distribution model with a relatively new method, namely the ISORM. Although the IFORM is recommended in the IEC standard, it has been shown to underestimate the probability behind the

surface of considered design variables (Chai and Leira, 2018; Dimitrov, 2020). This choice should therefore lead to more accurate values of the 50-year events. It should be noted that using the ISORM instead of IFORM results in a more conservative gust variable surface and a larger, or more extreme variable surface.





According to the probabilistic gust model derived in this analysis, the IEC ECD gust is conservative. That is, with the combination of coherent gust variables, the IEC gust is more extreme and falls outside the 3D gust parameter surface. By

increasing the return period to 460 years, the IEC gust is matched on the surface of the probabilistic gust surface. Though we did find that the IEC gust is more conservative with regards to the gust variables, it is interesting to see that higher loads can be reached with simulations from the probabilistic gust model. By including variability in the gust parameters, we were able to identify critical regions on the gust parameter surface that lead to loads that are significantly higher (up to 74%) that those that come from the simulation of the IEC ECD gust.

The current analysis is based on data from a single site, and is therefore likely to be site specific. Although previous studies on coherent structures (e.g. Hannesdóttir and Kelly, 2019; Belušić and Mahrt, 2012) imply that these kind of events occur at many different sites, the current probabilistic gust model is based on one site only. Future work could be to derive a more general gust model that would represent any site, or could be made site specific with IEC wind turbine class parameters.

We also note that all the results shown in Figures 5 and 6 are dependent on the DTU 10 MW wind turbine model, the DTU

basic controller, and the yaw controller used in the simulations. We do expect that commercial controllers designed by wind turbine manufacturers are more advanced and may react to extreme events in a different way. It would be possible to change the overall picture of the load results by tuning or optimizing the controller to handle gust cases with extreme gust accelerations, but that is outside the scope of the current study.

Although not discussed in the current work, the aerodynamic response under yawed inflow conditions is another topic of

concern since the accuracy of the blade element momentum method (BEM) decreases as the yaw error increases and generally such results should be considered carefully. However, the BEM implementation used in HAWC2 employs a yawed inflow correction and assumes non-constant induction across an angular ring element (see (Madsen et al., 2020)).

## 7    Conclusions

In this work observations of coherent gusts are used to obtain an environmental surface with a 50-year return period with

the Nataf distribution model. The surface is in three dimensional gust variable space of; rise time, amplitude and direction change. There is a large variability within the modeled gust variables, where the direction change and amplitude may exceed the ECD values, though in these cases with a considerably longer rise time. For modeled gusts with 10 s rise time, the maximum amplitude is 8.3 m/s and the maximum direction change is 35.3°

The modeled gust variable surface can match the values of the IEC ECD gust parameters by using a return period of

approximately 460 years.

We choose 3219 points on the surface to simulate for wind turbine response, where the simulations are performed with-, and without a yaw controller that is specially developed in this study.

The effect of the yaw controller is seen for the modeled gusts, where we see that the absolute maximum of all the considered load responses is lowered by including the yaw controller. Especially for gusts with a relatively long rise time where the blade

root torsion- and flap-wise bending moments are significantly reduced when the wind turbine yaws .



From the comparison of the modeled gusts and the IEC ECD, we find that even though the modeled gusts are not as severe in terms of gust variables, the difference in observed extreme loads are higher. From the considered load component channels the largest difference is seen for the tower base resultant moment, which is 74.2% higher for the modeled gust compared with the IEC gust. Similarly, the blade root torsion load is 63.9% higher for the modeled gust, and the blade root flap wise-, edge

wise, and the tower top yaw moments are around 20% higher than the simulated ECD gust. The maximum loads for $TB_{res}$, $BR_{flap}$, $BR_{edge}$, $BR_{torsion}$, and the $TT_{acc}$ are observed for the modeled gusts with short rise times, between 4.0 and 4.4 s, and low direction change between 14.9° and 21.6°, while the maximum load for the $TT_{yaw}$ is observed from a modeled gusts with a longer rise time of 186.2 s and a high direction change of 85.3°.

*Code and data availability.*    The data with the coherent gust variables and a Python code with the probabilistic gust model are available upon

request to the main author.





## Appendix A: Time series of the extreme cases for a selection of channels

**Figure A1.** Gust for which the tower bottom resultant bending moment $TB_{res}$ and the tower top acceleration $TT_{acc}$ has the absolute maximum (see also table 2)

**Figure A2.** Gust for which the tower top yawing moment $TT_{yaw}$ has the absolute maximum (see also table 2)





*Author contributions.* ÁH made the gust analysis and gust model. AU developed the yaw controller. DV performed HAWC2 load simulations. ÁH wrote the main body of the paper. DV and AU contributed with the writing of sections 4, 5, 6, and 7. All authors commented on the paper.

*Competing interests.* DTU Wind Energy develops, supports and distributes HAWC2 on commercial terms.

*Acknowledgements.* ÁH would like to thank Mark Kelly for comments and discussion around the coherent gust distribution model.



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
