# Peer review of "Extreme coherent gusts with direction change - probabilistic model, yaw control and wind turbine loads"

_Wind Energy Science, 2022_

## Author Response (AR1)

Dear reviewers,

We would like to express our sincere gratitude for carefully reading and reviewing our manuscript. Thank you for your constructive feedback and valuable suggestions.

We have carefully considered all the reviewer comments and updated our manuscript accordingly. We believe that the changes and updates based on your suggestions have greatly improved our manuscript.
Below you find the reviewer comments in black and our response in blue.

Reviewer 1 comments:

**Overall comments:**

- The manuscript presents results from a probabilistic model of a dataset consisting of 92 measurements of wind conditions meeting the authors' definition of a coherent gust. The probabilistic model uses the Nataf model to create a trivariate distribution of the data in terms of rise time, direction change, and amplitude change of the gust and ISORM is used to calculate environmental surfaces (i.e., combinations of these three variables with a common mean return period). The coherent wind gusts associated with the 50-yr environmental surface are then analyzed using a model of the DTU 10MW wind turbine in HAWC2 and the results are presented in terms of several cross-sectional demands on the tower and blades of the turbine. Simulations are implemented with and without a yaw controller. Results are compared with demands calculated following the definition of a coherent gust in IEC 61400-1.
- The authors present a clear and interesting analysis of an influential load case in the design of wind turbines. Their interpretation of the analysis provides useful insight and I enjoyed reading the manuscript. I have a few recommendations to increase clarity and accuracy, but otherwise recommend this paper for publication.

**Specific comments:**

- Line 6, the 460 year return period is written as if this is a general finding. Please revise to clarify that this return period is specific to a particular dataset at a particular location following a particular methodology.

  We have added text to the abstract, describing where the measurements are from and that the gust model is site specific.

- Line 16, DLCs not DLC's    Thanks. It is fixed
- Line 18, pls replace the wording "which is the target reliability level in wind turbine design." with something like "which is the intended recurrence period of the environmental conditions prescribed by the IEC Standard." The target reliability level depends on load and resistance factors in addition to the intended recurrence period of the environmental conditions of the DLC.

  That is right. We have changed the wording according to your suggestion.

- Lines 61-68, while the authors refer to a previous study for the detection and characterization of the coherent gusts within their dataset, it would be helpful to

provide a sentence or two here describing the criteria for an event to be categorized as a gust and the way in which the three variables are calculated for each event. In particular, please provide brief information on the spatial criteria for classification as a gust and the time average used to calculate the wind speed.

*Good point. We have added a few sentences describing the detection and characterization method.*

- Lines 68-69, how are the wind speeds Ua and Ub averaged in time and in space? I imagine this is described comprehensively in the authors' previous work, but it would help to provide some brief information here.

*The information was added in the description provided in the point above.*

- Section 3.1, the discussion on IFORM and ISORM needs revision. On Line 88, the statement that IFORM has been shown to underestimate the exceedance probability by an order of magnitude needs conditioning, as this is not a statement that is generally true. Can the authors elaborate what they are talking about? When multiple variables are being modeled probabilistically, there are many, equally legitimate, ways to determine exceedance probabilities and recurrence periods for combinations of these variables, so I am having trouble understanding what the authors mean when they say the exceedance probability is underestimated – underestimated compared to what?

*We agree that this needs revision. In the works of Dimitrov (2020) and Chai and Leira (2018) it is showed that with IFORM, only a fraction of the probability space behind the environmental curve/surface is considered when defining the exceedance probability in standard normal space. In our case, we want the exceedance probability to account for all events outside the environmental surface for the whole range of environmental parameters. Using IFORM in that case would lead to a non-conservative result. We have added an explanation of this in the manuscript and a new citation (Mackey and Haselsteiner, 2021) that explains very well the difference between ISORM and IFORM in terms of exceedance probabilities.*

- On Line 90, the authors refer to Equation (7) to distinguish between IFORM and ISORM, however this equation, which defines a sphere with radius beta in standard normal space, is used for both IFORM and for ISORM. The difference between the methods is in how the sphere is used to define the space of variables for which probability is calculated. In IFORM, the space is defined by a plane tangent to the sphere. In ISORM, which is a new method to me, I believe the space is defined as all points outside of the sphere. Can the authors explain more clearly the differences between IFORM and ISORM given that both require use of Equation (7)?

*You are right, and we meant to refer to equation 6 in line 90. This has now been corrected and we have added how beta is defined with IFORM in a footnote.*

- On Line 91, the authors refer to an exact solution for the return period using ISORM. Perhaps I am not understanding the authors' intent here, but I don't understand the idea of an exact solution for calculating a return period for combinations of three variables. Since multiple variables cannot be ranked unambiguously, there are many

ways to calculate exceedance probability/return periods for combinations of these variables. IFORM is one way. ISORM is another way. I don't think it's appropriate to call either one exact. They are just different. Are the authors saying that environmental surfaces using ISORM lead to more accurate calculations of probability of structural failure? If so, this should be clarified. And, even still, calling the result generally exact is too strong of a statement since this could only be true for a specific and simple idealization of the loading given the environmental variables.

We agree that the wording is of an exact solution is too strong and have changed it.

- Section 3.1.2, suggest editing the section title to emphasize IEC, e.g. "the IEC ECD" instead of "the ECD"

Thank you, we have added your suggestion.

- The result on Line 160 should be emphasized as being calculated for one specific site using one specific methodology.

We have added in the text in section 3.1.2 that our gust model is site specific and only based on measurements from Høvsøre.

- Line 232, bending not binding. Thank you. It is fixed
- Line 251, at this point the meaning of "load channels" was not clear to me. I eventually figured it out after seeing Table 2. It could be helpful to define this term earlier.

Good point. A definition has been added.

- Line 257, pitch not pith.  Fixed
- Line 271, as I was thinking about the results for TT_yaw, I was curious how much of the loading is inertial as the yaw controller accelerates the rotor. On a related note, does the yawing of the spinning rotor during an ECD cause a significant gyrotorque? This may be outside of scope, but, it my opinion, it would be interesting to provide some discussion on the influence of inertial loading compared to aerodynamic loading for this condition

It would indeed add an interesting discussion. It can be seen in figure A2 in the appendix that yawing of the spinning rotor does cause significant loading on the TT_yaw, where the absolute maximum loading occurs around 150 seconds after the gust has passed and the turbine is yawing. We think that this could be a topic of its own and for the sake of simplicity of our current paper we consider this discussion outside the topic.

- Table 2, suggest dropping a couple of significant figures from the reported moments.

A good suggestion that we have implemented.

Reviewer 2 comments:

General comments:

This paper is both relevant and well written and it contains material relevant for important discussion on how to improve modeling related to gust events. It contains a study on the actual wind conditions that is close to a ECD event as well as a consequence study of the measured windconditions simplified in to a parametric study of standardized IEC gust of the ECD type with variation in the parameters wind speed, wind direction and rise time.

The first part of the study that relates to the measured wind event is in general very good. Only minor remarks to this part to enhance the clarity. Eg how is it ensured that the selected wind events are related to a situations like an ECD, where the increased wind speed and wind direction stay on this new plateau for substantial amount of time? How is it ensured that the wind structure is coherent over a area covering a multi MW wind turbine? Perhaps this is covered in a prior reference, but it could be made more clear to the reader.

We agree that more explanation is needed here. We have added more text about the method and the scale of the observed events. The ECD like events at were observed in Høvsøre at two masts that were separated by 400 m.

It is also surprising that gusts with rise time down to 5s is included in this stude, as it is based on a previous study (Hannesdottir and Kelly, 2019) where the fastest gust seen for this site is 9s. How can this be? is this an artifact of the ISORM approach or can such fast events be justified as a coherent gust opposite being part of turbulence?

 The short rise time is indeed an artifact of the ISORM that is based only on the statistics of the events. When the statistics are extrapolated to 50-years, the rise time can become very short. We had some consideration about the shortest possible turnover time for a coherent fluctuation across the DTU 10 MW (p. 10 line 224). But a good direction for future work would be to include the scale considerations in the probabilistic gust model.

The second part of the paper addressing the consequence study of the chosen distribution of gust parameters is quite clear, and a public available turbine model with controller is used. It is nice that a public available turbine model is used as it makes it possible for other to reproduce the results. However, to conclude that the results found is what is expected from using an inductrial controller is not clear at all and the author should be cautious about concluding anything general from this analysis. From the time series shown in the appendix, it appears as the controller is highly sensitive to rapid changes in wind speed as well as highly sensitive to yaw errors which is understandable as there are nothing done in the controller to ensure low loads in such extreme situations. At most, conclusions from the consequence studies can be seen as indicative and not representative of the reponse of all wind turbines.

We completely agree with the reviewer that our results are highly dependent of the use of the DTU basic controller, and we do expect the results to change if a commercial controller was used. We have already stated this (p16 line 309): "We also note that all the results shown in Figures 5 and 6 are dependent on the DTU 10 MW wind turbine model, the DTU basic controller, and the yaw controller used in the simulations. We do expect that commercial controllers designed by wind turbine manufacturers are more advanced and may react to extreme events in a different way"

But to emphasize this further, we have added the word "highly" in front of "dependent" and restated this in the conclusion. Additionally, we added a future work section in which we acknowledge this needs to be addressed in a follow-up study.

Specific comments:

- Are the measured gust mainly related to the western or eastern sector. Is is mainly onshore ore offshore/near shore conditions that result in the measured gusts. Is there a difference in gusts from East or West?

Here all wind directions were included, except for filtering out the wake influenced sectors. The coherent gusts manly come from the predominant wind directions and from sectors where high winds are observed. It was shown in Hannesdóttir and Kelly (2019) that the ramp-like structures were also observed in Ryningsnäs and Østerild, displaying similar statistics at all sites.

- p.3 line 60. Is there a relation between start wind speed rise [m/s] and wind direction error as used in the standard, or how is it on this site?

There is a relation between start wind speed and direction change that can be seen in Hannesdóttir and Kelly (2019). It follows the same general trend as in the IEC standard, but we do not take this relation into consideration in our study as we only simulate events starting at 10 m/s.

- p.3 line 66. "from a variety of phenomena" like what? Can you give examples to the reader?

We have added some examples of phenomena in the text.

- How are the wind measurements done? Height of measurement, number of points, single or multiple metmast? Does a rise time of 5 seconds still correspond to a coherent gust with a spatial size of +100m?

We have added information about measurement heights, frequency, masts and more at the beginning of section 3. Regarding the rise time, see discussion above.

- How is it ensured that eg the wind direction is "permanent" and not just a temporary gust returning at a low value after short time? Same question for the delta wind speed.

A better explanation of the detection method has been added. But then nothing is "permanent" in the atmosphere and all these events are a part of large wind speed fluctuations. Usually, the wind speed stays constant for some time or decreases much more slowly than the observed wind speed increase (Hannesdóttir et al. 2019).

- How is it ensured that gust structures are large enough to quality for a coherent structure for a multi-MW turbine?

The detected events are coherent in the sense that they are observed at all measurement heights (60m, 100m and 160m) and observed at 2 masts separated by 400 m. This information has been added to the text.

- It is stated that 92 gusts have been detected. How is it ensured that there is enough points?

We assume we have a large enough sample, as we have a good fit of the theoretical distributions to our data points. It is difficult to say when the number of points is too low, as it might depend on the distribution of the data points.

- p.5 line 115. What is IDF?

This should have been ISORM and has now been corrected.

- p.6 line 132. it is unclear how the correlation coefficients rho_ij calculated?

We have added an explanation.

- p.6 line 137. How is U derived?

U = (u_1, u_2, u_3). This has been added to the text.

- p6 line 138. Is equation 12 to be understood as a dot product?

Matrix multiplication. We removed the dot

- p.7 Figure 2. It is quite difficult to see any quanticative results of these plot. In the right plot is appears as not data a present for a delta u<10m/s, whereas the left plot show the majority of points below 10m/s. Perhaps the points can be placed in Figure 3 with colors representing the rise time.

Good suggestion. We have changed Figure 3 accordingly and made the points more visible in Figure 2.

- p.7 line 156. Why is the rise time negative here?

We changed this

- p.8 eq(17) Why is this shown as a dot product?

Matrix multiplication. We removed the dot

- p 10. It would be nice if eg the tower top resulting bending moment was included as well.

Just as also reviewer 1 pointed out that it would be interesting and relevant to further extend the discussion regarding intertial loading, we suggest to refer the discussion on the more detailed analysis of how the yaw controller can further influence the response to future work.

- p 10. Line232. "binding" -> "bending"

Thanks, fixed now.

- p11. line 246. "may be seen". Can it be seen or can it not be seen? Please choose.

They can be seen

- p 11. Fig 5 Please write in captions what sensor is seen in the plot and or make it more clear from the individual figures

We extended the caption and hopefully this made the figures more clear. Note that the channel is indicated as the vertical title right of the colorbar.

- p 12. Fig 6 Please write in captions what sensor is seen in the plot and or make it more clear from the individual figures

Same as above.

- p 16. Line 315. Are you sure the accuracy to the BEM model in HAWC2 decrease with yaw error? otherwise the work "may" should be included beween "(BEM)" and "decrease"

Good point. Generally speaking, the induced velocities as calculated by the HAWC2 BEM method are not necessarily inaccurately calculated at high yaw errors (see for example figures 8 and 9 from https://www.wind-energ-sci.net/5/1/2020/). However, as the yaw error increases the angle of attack variations over one revolution of the rotor will increase as well. Consequently, at larger yaw angles part of the blade will transition from attached to stalled flow. When operating under stalled conditions there is an incrased uncertainty since the dynamics of the stalled 3D flow are not accurately modelled in HAWC2 BEM, but this uncertainty is not exactly well defined or quantified in this work so I agree adding a "may" is appropriate. We also added that this (BEM based blade loads for very large yaw errors) could be investigated in more detail in future work.

- p16. Line 316. Is is fine to reflect on the model accuracy, but as I read the paper, I am more concerned about the uncertainty in the simple turbine power controller than the aerodynamics.

That is a good comment. The work we present is indeed much more focussed on the statisticall modelling and the general response of an existing, simplified academic controller. I think you also might be pointing at the response we can see in figure  A1 that there could be room for improvement considering how much the pitch controller is overshooting the response and how that is driving the loading of the tower. We did consider addressing this more promentally in the paper in earlier internal revisions but ended up deciding against it so we could limit the scope of this publication and focus more on the general response and the statistical model. However, in the discussion (page 16, lines 309-313) we do aknowledge that in general the controller can change the picture of the response significantly. We have now also added this the conclusions and to future work.

- Appendix A. Figure A1 Please include wind speed and wind direction for clarity.

Thanks for the good suggestion, added.
The dottet line is for the IEC ECD gust for reference, the solid line is the specefic gust case indicated in the title. The label explaining this is only repeated in the upper left figure. I have added this to the caption to make this more clear.

- Appendix A. Figure A1. Gust parameters are written in the figure title, but it is unclear what the numbers represents.

Agreed, thanks for pointing this out. I have added to the title that they refer to the gust amplitude, rise time and direction change.

- Appendix A. Figure A2. See comments related to figure A1

See above.

---

## Author Response (AR2)

Dear Julie,

Thank you for editing our paper and providing comments for improvement.
We have implemented your suggestion by removing the Future work section and instead added a paragraph at the end of the Conclusion section with considerations for future work.

We have also provided a DOI for HAWC2 files: 10.11583/DTU.21717524 under 'code and data availability'. It will become active after the paper is published because one of the fields in this item must link to the DOI of the actual publication. There is a draft link to the DOI at: https://figshare.com/s/4601e4ef23a0e1d020bc

Best regards,
Ásta, also on behalf of the co-authors